# HideMIA: Hidden Wavelet Mining for Privacy-Enhancing Medical Image Analysis

Xun Lin*
Beihang University
linxun@buaa.edu.cn

Yi Yu*
Nanyang Technological University
yuyi0010@e.ntu.edu.sg

Zitong Yu[†]
Great Bay University
yuzitong@gbu.edu.cn

Ruohan Meng
Nanjing University of Information
Science and Technology
ruohanmeng.melody@gmail.com

Jiale Zhou
Beihang University
zhoujiale@buaa.edu.cn

Ajian Liu
Institute of Automation, Chinese
Academy of Sciences
ajian.liu@ia.ac.cn

Yizhong Liu
Beihang University
liuyizhong@buaa.edu.cn

Shuai Wang
Beihang University
wangshuai@buaa.edu.cn

Wenzhong Tang
Beihang University
tangwenzhong@buaa.edu.cn

Zhen Lei
MAIS, CASIA
School of Artificial Intelligence, UCAS
CAIR, HKISI, CAS
zlei@nlpr.ia.ac.cn

Alex Kot
Rapid-Rich Object Search Lab (ROSE),
Nanyang Technological University
eackot@ntu.edu.sg

## Abstract

Despite the advancements that deep learning has brought to medical image analysis (MIA), protecting the privacy of images remains a challenge. In a client-server MIA framework, especially after deployment, patients' private medical images can be easily captured by attackers from the transmission channel or malicious third-party servers. Previous MIA privacy-enhancing methods, whether based on distortion or homomorphic encryption, expose the fact that the transmitted images are medical images or transform the images into semantic-lacking noise. This tends to alert attackers, thereby falling into a *cat-and-mouse game* of theft and protection. To address this issue, we propose a covert MIA framework based on deep image hiding, namely HideMIA, which secures medical images by embedding them within natural cover images that are unlikely to raise suspicion. By directly analyzing the hidden medical images in the steganographic domain, HideMIA makes it difficult for attackers to notice the presence of medical images. Specifically, we propose the Mixture-of-Difference-Convolutions (MoDC) and Asymmetric Wavelet Attention (AsyWA) to enable HideMIA to conduct fine-grained analysis on each wavelet sub-band within the steganographic domain, mining features that are specific to medical images. Moreover, to reduce resource consumption on client devices, we design function-aligned knowledge distillation to obtain a lightweight hiding network, namely LightIH. Extensive experiments on six medical datasets demonstrate that our HideMIA achieves superior MIA performance and protective imperceptibility on medical image segmentation and classification.

## CCS Concepts

• **Computing methodologies → Computer vision problems**.

## Keywords

Image hiding, medical image analysis, privacy-enhancing

**ACM Reference Format:**
Xun Lin, Yi Yu, Zitong Yu, Ruohan Meng, Jiale Zhou, Ajian Liu, Yizhong Liu, Shuai Wang, Wenzhong Tang, Zhen Lei, and Alex Kot. 2024. HideMIA: Hidden Wavelet Mining for Privacy-Enhancing Medical Image Analysis. In *Proceedings of the 32nd ACM International Conference on Multimedia (MM '24), October 28–November 1, 2024, Melbourne, VIC, Australia.* ACM, New York, NY, USA, 10 pages. https://doi.org/10.1145/3664647.3680806

*Both authors contributed equally to this paper. [†]Corresponding author.

## 1 Introduction

Deep learning technologies have rapidly advanced in Medical Image Analysis (MIA) and have become an integral part of modern medical diagnostics [19]. These technologies significantly improve the accuracy and efficiency of clinical diagnosis, *e.g.* early screening for major diseases such as cancer [31]. Despite the impressive performance of deep neural networks, there are increasing concerns about the security issues [30, 39, 49, 50, 56–58] associated with artificial intelligence. Among these security issues, the protection of patient privacy remains a severe challenge [38]. From a legal and ethical perspective, patients' medical images must be rigorously protected, ensuring the security and privacy of these images during their storage, transmission, and analysis processes

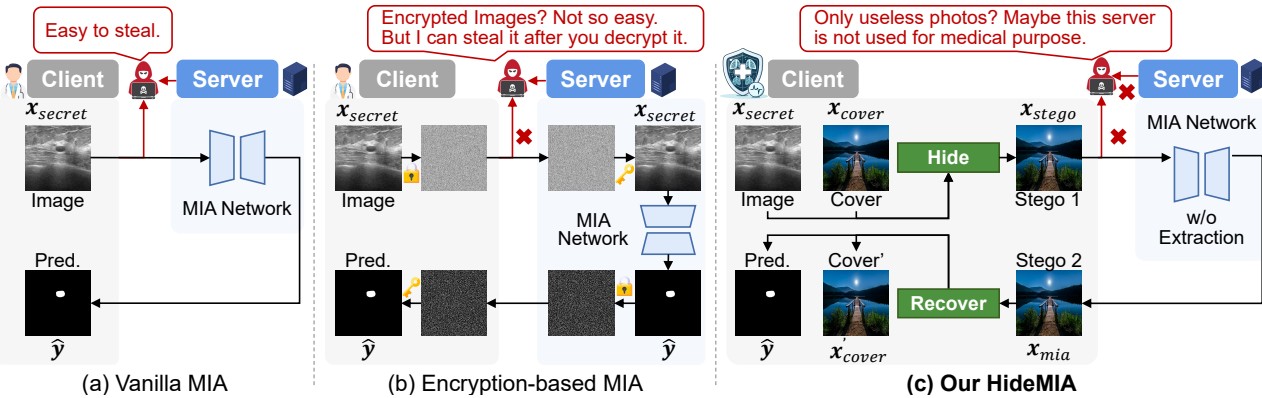

**Figure 1: Different client-server frameworks for MIA, including (a) vanilla, (b) encryption-based, and (c) the proposed HideMIA framework. Our HideMIA makes the attacker hardly notice the existence of medical images.**

[18]. Some medical institutions have adopted methods such as data anonymization or pseudonymization [3] to mitigate the risk of data leakage. However, real-world scenarios indicate that these methods are not robust against re-identification attacks [19]. As illustrated in Fig. 1(a), there is a risk of leakage, especially during the transmission of images from a client to a server [2]. Although techniques like non-homomorphic image encryption can prevent image leakage during transmission (see Fig. 1(b)), they are not robust to attacks from malicious servers [20]. Once an attacker controls the server by system vulnerabilities or embedded malicious backdoors (as a third-party server provider), the decrypted images on the server-side are also at the risk of being captured.

Currently, most image privacy enhancement technologies focus on the training phase, for instance, by enabling different institutions to collaboratively train models without the direct exchange of image data through federated learning [11], or by generating unlearnable examples to prevent unauthorized model training [30]. These methods effectively enhance the privacy protection of training data, yet image protection after the model deployment remains a challenge. Some studies focus on protecting images after deployment through distortion-based [9, 20, 40] methods to modify detailed information of the original images. Although these methods somewhat protect image privacy, attackers can easily notice that the transmitted images are medical images. To further enhance security, recent works propose encoding-based [21, 45] and homomorphic-encryption-based [55, 63] methods to protect medical images. However, these methods significantly alter the distribution of the images, easily alerting attackers to the fact that the images are protected by specific techniques, thus falling into a *cat-and-mouse game* of encryption and decryption. Among these methods, methods based on homomorphic encryption are time-consuming, making them impractical for many real-world applications.

Recent progress in deep image hiding (DIH) networks [12, 17, 26], which are proposed to conceal a secret image within a cover image for covert image transmission, has inspired us to propose a novel framework to perform **Hid**den **M**edical **I**mage **A**nalysis (**HideMIA**) based on DIH. As shown in Fig. 1(c), HideMIA hides a medical image within a natural photo, allowing for direct MIA in the steganographic domain without the image extraction and decryption on the server-side. However, directly performing MIA

in the steganographic domain is challenging because the medical-specific features within stego images are subtle, and the conspicuous information from the cover images introduces noise to the MIA.

Considering that DIH networks tend to hide the information of the secret image in high-frequency parts of the cover images [17], we design the spectral-aware **M**ixture-**o**f-**D**ifference-**C**onvolutions (**MoDC**) and **Asy**mmetric **W**avelet **A**ttention (**AsyWA**) to solve the problems above. These two modules are integrated with HideMIA's server-side MIA network. They can perform band-adaptive analysis on stego images in the steganographic domain (targeting different bands after discrete wavelet transform [14]). Specifically, within MoDC, inspired by [23, 29, 46, 59], we introduce various novel pixel difference convolutions to extract features of the hidden medical images in a fine-grained manner. Considering that the content and form of hidden information may vary across different bands, we bring the idea of the Mixture-of-Experts (MoE) [24, 62] to these convolutions, enabling adaptive combinations of these difference convolutions for each frequency band. Meanwhile, we propose inter-band and inner-band cross-attention with AsyWA to enable a global perception of concealed medical images across spatial and spectral dimensions. Considering that most of the semantic information in the cover belongs to the low-frequency bands, we design an asymmetric interaction constraint for inter-band cross-attention to ensure global perception while preventing cover-specific features in the low-frequency band from hindering the extraction of medical-related features in the high-frequency bands.

Moreover, existing DIH networks are not resource-friendly for clients with limited computing resources, *e.g.,* medical imaging devices. Therefore, motivated by [33], we propose the function-aligned knowledge distillation to obtain a **light**weight D**IH** network, namely **LightIH**. Unlike the feature-aligned knowledge distillations, our function-aligned distillation ensures that the student network learns sensitive features that influence analysis performance and imperceptibility during the distillation process. We employ the widely-used INN-based and spectrum-aware DIH network, HiNet, which is sensitive to manipulations in the steganographic domain [8], as the teacher network. To the best of our knowledge, this is the first work to develop a lightweight DIH network through knowledge distillation.

Our contributions can be summarized as follows:

• We propose a privacy-enhancing client-server MIA framework, namely HideMIA, to covertly analyze medical images without raising the suspicions of attackers.

• We design MoDC and AsyWA for wavelet-transformed stego images to ensure that HideMIA can accurately perform MIA in the steganographic domain without interference from conspicuous information in the covers.

• We develop LightIH, which, through the proposed function-aligned distillation, reduces the resource consumption of the DIH network while enhancing the accuracy of the MIA process.

• Extensive experiments on three image segmentation datasets and three image classification datasets demonstrate the effectiveness and imperceptibility of HideMIA.

## 2 Related Works

### 2.1 Privacy Enhancement for Medical Images

**Distortion-based Methods.** To prevent threats from transmission channels and malicious servers, privacy-preserving client-server frameworks are proposed for medical image analysis. Kim et al. [22] protect the privacy of medical images in the segmentation pipeline by adding the images sent from the client with reference images. Kim et al. [20], Packhäuser et al. [40] propose deformation generators to produce pseudo-random non-linear deformation for images from the client, which allows both distortion and recovery of the sensitive information of medical images and segmentation results. These distortion-based methods can defend against reconstruction and re-identification attacks, however, they still let attackers recognize that the transmitted images are medical.

**Encoding-based and Encryption-based Methods.** To further enhance privacy, encoding-based and encryption-based are proposed to change the content distribution of medical images. [21] design an encoder to remove identity-related information from medical images and propose a discriminator to identify ROI. Shiri et al. [45] adopt a learnable auto-encoder that employs convolution operations for the sparse transformation of medical images and adds pseudo-random noise to further obfuscate them. The effectiveness of homomorphic encryption is discussed in [55, 64] to further improve privacy protection for distributed medical image segmentation. Although encoding-based and encryption-based methods achieve better protective performance, these methods alter the image into feature maps or noise that are difficult for humans to understand. This may easily alert attackers and prompt them to design more threatening attacks.

### 2.2 Deep Image Hiding

Image hiding aims to covertly conceal a secret image within a cover image and enables the extraction of the hidden secret image. Baluja [4], Hayes, and Danezis [13] are the first to propose DIH networks based on the encoder-decoder structure. Liu et al. [36] improve the encoder based on U-Net and discrete wavelet transformation (DWT) [14] to embed the secret image, further enhancing the reversibility. Recent progress on invertible neural networks (INNs) in various image-to-image tasks [51, 52] has inspired the application of INNs in image hiding. Lu et al. [37] design INN-based image hiding by modeling the embedding and extraction processes as the forward and inverse operations in affine transformations. To enhance reversibility and imperceptibility, Deng et al. [8, 12, 17] input the wavelet-transformed cover image and secret image into an INN-based image hiding network, thereby embedding the secret into the high-frequency components of the cover. Xu et al. [53] propose a conditional normalizing flow to model the distribution of the redundant high-frequency component conditioned on the cover images, enhancing robustness against distortion.

## 3 HideMIA

### 3.1 Overall Framework

Our HideMIA consists of image hiding network $\mathcal{H}(\cdot, \cdot)$, medical image analysis network $\mathcal{M}(\cdot)$, and image recovering networks $\mathcal{R}(\cdot)$. The image hiding and image recovering networks are deployed on the client-side, while the medical image analysis network operates on the server. Let $x_{secret}$ denote the image captured by medical imaging devices, and $x_{cover}$ represent the cover image (**which can be any natural image**) adopted to conceal $x_{secret}$ within. The process $x_{stego} = \mathcal{H}(x_{secret}, x_{cover})$ yields a stego image, wherein $x_{secret}$ is embedded into $x_{cover}$. After the concealment, $x_{stego}$ is transmitted from the client to the server, where it is directly fed into the MIA network: $x_{mia} = \mathcal{M}(x_{stego})$. The MIA network analyzes the concealed medical image within the steganographic domain. Upon returning $x_{mia}$ to the client, the recovery network extracts the hidden analysis results as $\hat{y} = \mathcal{R}(x_{mia})$. In this pipeline, both $x_{mia}$ and $x_{stego}$ are easily captured by attackers, thus we require $x_{mia}$ and $x_{stego}$ remain visually indistinguishable from $x_{cover}$ to ensure covert MIA without raising the attackers' suspicion.

### 3.2 MIA in the Steganographic Domain

We observe that existing MIA methods struggle to conduct precise and covert analysis within the steganographic domain. They often produce numerous false alarms and struggle to maintain visual consistency between $x_{mia}$ and $x_{stego}$, primarily due to the presence of conspicuous information in the cover image. Consequently, we propose the Mixture-of-Difference-Convolutions (MoDC) and Asymmetric Wavelet Attention (AsyWA) to perform fine-grained and cover-agnostic MIA in the steganographic domain. In our MIA network, $x_{stego}$ is decomposed into four wavelet sub-bands using DWT: $x_{stego}^{LL}, x_{stego}^{LH}, x_{stego}^{HL}, x_{stego}^{HH} \in \mathbb{R}^{\frac{H}{2} \times \frac{W}{2} \times 3}$. These represent low (L) and high (H) frequencies across horizontal and vertical directions, detailing $x_{stego}$'s frequency spectrum diversely. We utilize U-Net [44] as the backbone for each sub-network, with the output dimension segmentation heads set to 3. Both MoDC and AsyWA are integrated within each sub-network. The outputs from these sub-networks, namely $x_{mia}^{LL}, x_{mia}^{LH}, x_{mia}^{HL}, x_{mia}^{HH} \in \mathbb{R}^{\frac{H}{2} \times \frac{W}{2} \times 3}$, are combined using Inverse Wavelet Transform (IWT) to compose $x_{mia}$, and then returned to the client.

Besides, we notice the guidance of original $x_{stego}^{LL}$ can significantly improve the visual similarity between $x_{mia}$ and $x_{stego}$, thereby enhancing the HideMIA's imperceptibility. Specifically, since most of the semantic information in the cover, rather than the secret, is distributed within the LL band, we allow the LL sub-network to learn the residual of $x_{stego}^{LL}$ as follows:

$$x_{mia} = \text{IWT}(x_{stego}^{LL} + x_{mia}^{LL}, \ x_{mia}^{LH}, \ x_{mia}^{HL}, \ x_{mia}^{HH}). \quad (1)$$

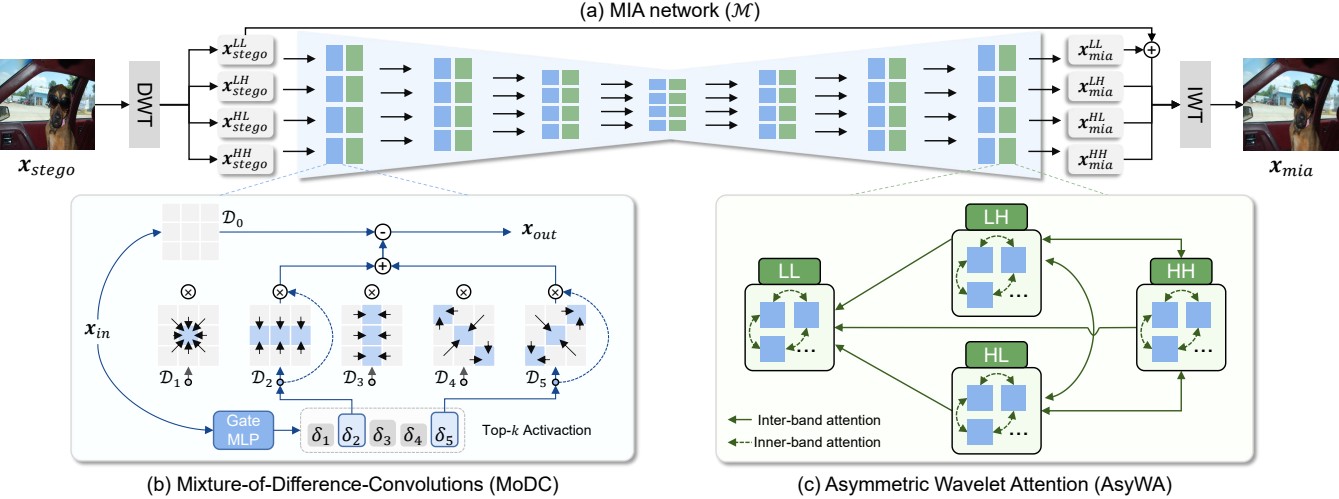

**Figure 2: Illustration of (a) the server-side MIA network of the proposed HideMIA, (b) MoDC, and (c) AsyWA.**

This operation has little impact on MIA performance and significantly improves HideMIA's imperceptibility (even feasibility). In the following sections, we provide detailed descriptions of the proposed MoDC and AsyWA.

**Mixture-of-Difference-Convolutions.** To ensure imperceptibility and recoverability, DIH networks tend to conceal different components of the secret image within different frequency bands of the cover images in different manners [25]. This makes the thorough analysis of the concealed information within each band using the same type of convolution operation challenging. Drawing inspiration from Central Difference Convolution (CDC) in detail-required vision tasks (*e.g.,* face anti-spoofing [59]), we propose four novel directional difference convolution operations, *i.e.,* horizontal, vertical, left diagonal, and right diagonal, to adapt to different bands generated by DWT, accommodating the high- and low-frequency bands in horizontal and vertical orientations.

Additionally, inspired by the idea of Mixture-of-Experts (MoE) [32, 62], we design a novel structure based on the newly introduced directional difference convolution operations, namely MoDC. As shown in Fig. 2(b), MoDC enables the network to adaptively combine the differential convolution operations that are most apt for each frequency band. Specifically, the vanilla convolution $\mathcal{D}_0$ and the aforementioned five difference convolutions, *i.e.,* $\mathcal{D}_1$ (central), $\mathcal{D}_2$ (horizontal), $\mathcal{D}_3$ (vertical), $\mathcal{D}_4$ (left diagonal), and $\mathcal{D}_5$ (right diagonal), can be described as follows:

$$
\begin{aligned}
\mathcal{D}_0(r_x, r_y) &= \sum_{(\Delta r_x, \Delta r_y) \in \mathcal{R}} w(\Delta r_x, \Delta r_y) \cdot x_{in}(r_x - \Delta r_x, r_y - \Delta r_y), \\
\mathcal{D}_1(r_x, r_y) &= \sum_{(\Delta r_x, \Delta r_y) \in \mathcal{R}} w(\Delta r_x, \Delta r_y) \cdot x_{in}(r_x, r_y), \\
\mathcal{D}_2(r_x, r_y) &= \sum_{(\Delta r_x, \Delta r_y) \in \mathcal{R}} w(\Delta r_x, \Delta r_y) \cdot x_{in}(\Delta r_x, r_y), \\
\mathcal{D}_3(r_x, r_y) &= \sum_{(\Delta r_x, \Delta r_y) \in \mathcal{R}} w(\Delta r_x, \Delta r_y) \cdot x_{in}(r_x, \Delta r_y), \\
\mathcal{D}_4(r_x, r_y) &= \sum_{(\Delta r_x, \Delta r_y) \in \mathcal{R}} w(\Delta r_x, \Delta r_y) \cdot x_{in}(\Delta r_x + \Delta r_y, \Delta r_x + \Delta r_y), \\
\mathcal{D}_5(r_x, r_y) &= \sum_{(\Delta r_x, \Delta r_y) \in \mathcal{R}} w(\Delta r_x, \Delta r_y) \cdot x_{in}(\Delta r_x - \Delta r_y, \Delta r_y - \Delta r_x),
\end{aligned}
\tag{2}
$$

where $\mathcal{R} = \{(1,1), (0,1), \cdots, (-1,0), (-1,-1)\}$ is the local respective field of the trainable 3×3 vanilla convolution kernel $w$, and $r_x, r_y$ denote the current position of the kernel conducting on $x_{in}$ and $x_{out}$. The Gate Multi-Layer Perception (MLP) [48] in Fig. 2 is composed of a sequence of layers: pooling, linear, GELU activation, another linear, and finally Softmax. Gate MLP receives the input feature $x_{in}$ and outputs the weights $\delta \in \mathbb{R}^5$ of five difference convolutions. MoDC then selects $P = \text{top}_k(\delta)$ kernels for activation and normalize their weights. This step is designed to prevent kernels that are not suitable for covert information analysis in the current band. Only the most appropriate difference kernels are conducted on the input features $x_{in}$. Subsequently, the outputs of the activated difference convolutions are weighted by their corresponding weights $\delta_i$, summed, and then subtracted from the output of the vanilla convolution kernel $\mathcal{D}_0$, completing the spectrum-aware difference convolution. MoDC is formulated as follows:

$$
x_{out} = \mathcal{D}_0(x_{in}) - \lambda \cdot \sum_{i \in S} \frac{\delta_i}{\sum_{j \in P} \delta_j} \cdot \mathcal{D}_i(x_{in}),
\tag{3}
$$

where $\lambda$ is used to trade-off between the intensity of vanilla convolution and difference convolutions.

**Asymmetric Wavelet Attention.** As mentioned above, different components of medical images are concealed in varying forms within different frequency bands of the cover image. Therefore, enabling inter-band interactions to acquire a band-wise global perception of the concealed medical images is important. Previous works [12, 17] on DIH validate that the secret image is primarily concealed within the high-frequency parts of an image, *e.g.* within the non-LL bands in wavelet-transformed images. Consequently, during inter-band interactions, the semantic information within the low-frequency band can easily interfere with the fine-grained analysis of other bands.

To solve this problem, we propose the AsyWA, which allows inter-band cross-attention [6] among the features of different frequency bands, achieving global-spectrum perception. Bidirectional feature interactions are permitted among the high-frequency bands (*i.e.,* LH, HL, and HH), but the LL band can only receive cues from

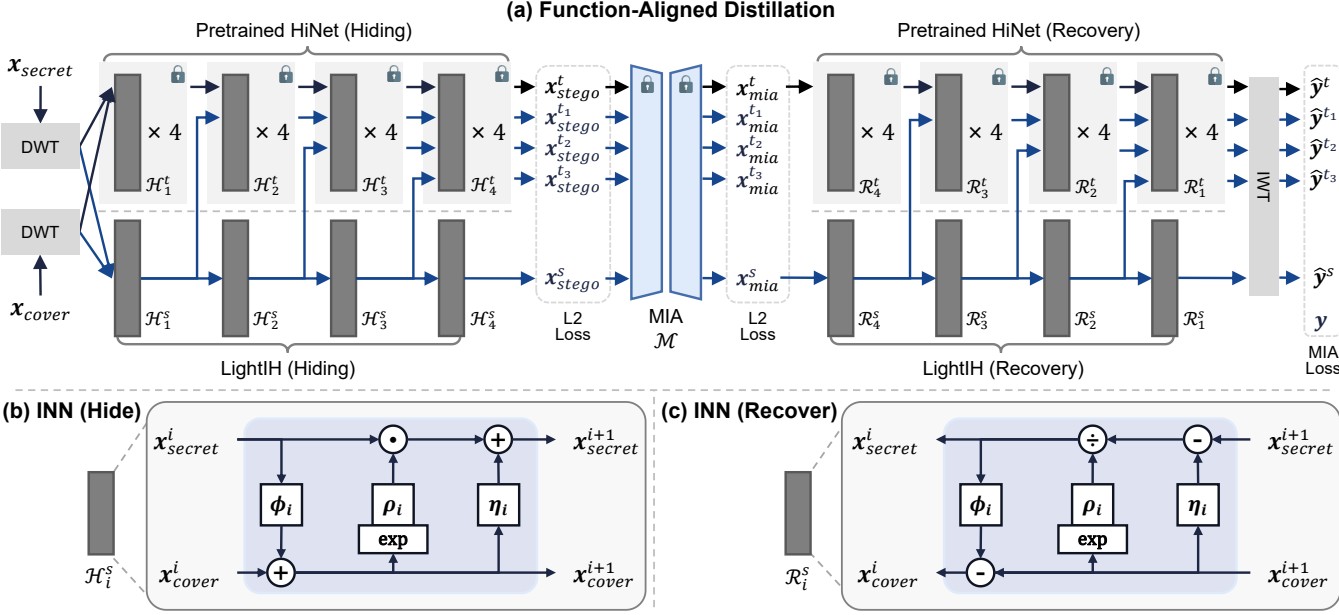

**Figure 3: Illustration of LightIH. (a) Function-aligned distillation. (b) Invertible neural network for image hiding. (c) Invertible neural network for image recovery.**

the high-frequency bands. AsyWA prevents the cover-specific semantic features mined from the low-frequency spectrum during MIA from misleading the extraction of the medical-specific fine-grained information. Additionally, considering the significance of spatial correlations between patches within the same band, we also integrate inner-band attention within AsyWA. Both our inter-patch and inner-band attention mechanisms are implemented based on multi-head attention [35]. AsyWA with inter- and inner-band attention can be formulated as follows:

$$\text{MHA}(\boldsymbol{x}_1, \boldsymbol{x}_2) = \frac{Q(\boldsymbol{x}_2)K(\boldsymbol{x}_1)^\top}{\sqrt{d}} V(\boldsymbol{x}_1), \tag{4}$$

$$\boldsymbol{x}_{out}^{LL} = \boldsymbol{x}_{in}^{LL} + \sum_{b \in B} \text{MHA}(\boldsymbol{x}_{in}^{LL}, \boldsymbol{x}_{in}^{b}), \tag{5}$$

$$\boldsymbol{x}_{out}^{h} = \boldsymbol{x}_{in}^{h} + \sum_{h' \in H} \text{MHA}(\boldsymbol{x}_{in}^{h}, \boldsymbol{x}_{in}^{h'}), \quad \forall h \in H, \tag{6}$$

where $B = \{LL, HL, LH, HH\}$ denotes the band set of DWT, $H = \{HL, LH, HH\}$ represents the high-frequency band set, $d$ denotes the number of pixels in the input feature, $Q, K, V$ are the linear projections corresponding to the *query*, *key*, and *value*, respectively.

## 3.3 Lightweight DIH Network

To make the $\mathcal{H}$ and $\mathcal{R}$ resource-friendly for clients, we perform knowledge distillation [60] on the state-of-the-art (SOTA) DIH network, HiNet [17], to obtain a lightweight network, namely LightIH. Please note that HiNet also integrates DWT to improve stego imperceptibility. Existing feature distillation methods for vision tasks commonly [10] use distance such as L2 to align the features at corresponding stages between the student and teacher networks [27]. However, within the HideMIA framework, which includes multiple parts, *i.e.*, $\mathcal{M}$, $\mathcal{H}$, and $\mathcal{R}$, changing the intermediate features in different directions by the same L2 distance at early stages can lead to big differences in the final predictions [34].

To this end, we propose a function-aligned distillation strategy, where we encourage feature alignment solely from the perspective of DIH and MIA performance. This means that the features or outputs from corresponding stages of the teacher and student networks, once fed into the latter part of the same network, should yield closely similar results. Our function-aligned distillation encourages the student to focus more on the sensitive directions concluded by the teacher that significantly impact HideMIA's performance, rather than overly emphasizing the replication of the teacher's intermediate outputs. To ensure HideMIA possesses better MIA capabilities and imperceptibility, we combine the distillation loss, visual consistency loss, and MIA loss to supervise LightIH.

As illustrated in Fig. 3, we adopt the pretrained HiNet (16 layers) as the teacher network. Our student network LightIH consists of 4 INN layers (a quarter of HiNet). We conduct the function-aligned distillation across all phases of HideMIA (hiding, MIA, and recovery). Specifically, both the teacher and student networks are divided into four stages. In the teacher network, each stage consists of 4 INNs, denoted as $\mathcal{H}_i^t$ for hiding and $\mathcal{R}_i^t$ for recovery, while in the student network, each stage includes only one INN ($\mathcal{H}_i^s$ and $\mathcal{R}_i^s$). Given that the two DIH backbone are INN-based, the convolution layer weights ($\phi_i$, $\rho_i$, and $\eta_i$) in $\mathcal{H}_i$ and $\mathcal{R}_i$ are shared. As shown in Figs. 3(b)-(c), only the sequence and operations applied to the inputs differ. Taking the hiding stage as an example, we feed the middle output from the student's $\mathcal{H}_i^s$ into the teacher's $\mathcal{H}_{i+1}^t$ and pass it through the subsequent blocks of the teacher to obtain $\boldsymbol{x}_{stego}^{t_i}$. As formulated in Fig. 3(a) and Eq. (7), these stego images and $\boldsymbol{x}_{stego}^s$ are then supervised with the teacher network's original output $\boldsymbol{x}_{stego}^t$ to guide the function alignment of image hiding.

Similarly, after passing through the MIA network, we use a comparable function-aligned supervision (see Eq. (7)). In the recovery

---

**Algorithm 1:** Training Process of HideMIA

**Input** : Teacher DIH networks $\mathcal{H}^t, \mathcal{R}^t$, student DIH networks $\mathcal{H}^s, \mathcal{R}^s$,
MIA network $\mathcal{M}$, training dataset $\mathcal{T}$, epoch numbers of stages 1, 2,
and 3: $e_1, e_2, e_3$, respectively, and learning rates $\ell_r^1, \ell_r^2, \ell_r^3$.

**for** $j \leftarrow 1$ **to** $e_1 + e_2 + e_3$ **do**
  **for** $\{x_{cover}, x_{secret}, y\} \in \mathcal{T}$ **do**
    $x_{stego}^t \leftarrow \mathcal{H}^t(x_{cover}, x_{secret})$;
    $x_{mia}^t \leftarrow \mathcal{M}(x_{stego}^t)$;
    $\hat{y}^t \leftarrow \mathcal{R}^t(x_{mia}^t)$;
    **if** $j \leq e_1$ **then**
      *# Stage 1: optimizing $\mathcal{M}$*
      $\mathcal{L}_{total} \leftarrow \beta_1 \cdot \mathcal{L}_{mia}(\hat{y}^t, y) + \beta_2 \cdot \mathcal{L}_{percept}(x_{mia}^t, x_{cover})$
      $\theta_{\mathcal{M}} \leftarrow \theta_{\mathcal{M}} - \ell_r^1 \cdot \nabla_{\theta_{\mathcal{M}}} \mathcal{L}_{total}$
    **else if** $j \geq e_1$ **then**
      *# Stage 2: using $\mathcal{H}^t$ and $\mathcal{R}^t$ to distill $\mathcal{H}^s$ and $\mathcal{R}^s$*
      $x_{stego}^s, \{x_{stego}^{t_i}\}_{i=1}^3 \leftarrow \mathcal{H}^s(x_{cover}, x_{secret}, \mathcal{H}^t)$;
      $x_{mia}^s, \{x_{mia}^{t_i}\}_{i=1}^3 \leftarrow \mathcal{M}(x_{stego}^s, \{x_{stego}^{t_i}\}_{i=1}^3)$;
      $\hat{y}^s, \{\hat{y}^{t_i}\}_{i=1}^3 \leftarrow \mathcal{R}^s(x_{mia}^s, \{x_{stego}^{t_i}\}_{i=1}^3, \mathcal{R}^t)$;
      $\mathcal{L}_{dist}^{hide} \leftarrow \ell_2(x_{stego}^s, x_{stego}^t) + \sum_{i=1}^3 \ell_2(x_{stego}^{t_i}, x_{stego}^t)$
      $\mathcal{L}_{dist}^{mia} \leftarrow \ell_2(x_{mia}^s, x_{mia}^t) + \sum_{i=1}^3 \ell_2(x_{mia}^{t_i}, x_{mia}^t)$
      $\mathcal{L}_{dist}^{recover} \leftarrow \mathcal{L}_{mia}(\hat{y}^s, y) + \sum_{i=1}^3 \mathcal{L}_{mia}(\hat{y}^{t_i}, y)$
      $\theta_{\mathcal{H}^s} \leftarrow \theta_{\mathcal{H}^s} - \ell_r^2 \cdot \nabla_{\theta_{\mathcal{H}^s}} (\beta_1 \cdot \mathcal{L}_{dist}^{hide} + \beta_2 \cdot \mathcal{L}_{dist}^{mia})$
      $\theta_{\mathcal{R}^s} \leftarrow \theta_{\mathcal{R}^s} - \ell_r^2 \cdot \nabla_{\theta_{\mathcal{R}^s}} (\beta_3 \cdot \mathcal{L}_{dist}^{recover})$
      **if** $e_2 \leq j \leq e_3$ **then**
        *# Stage 3: optimizing $\mathcal{M}, \mathcal{H}^s$, and $\mathcal{R}^s$*
        $\theta_{\mathcal{H}^s} \leftarrow \theta_{\mathcal{H}^s} - \ell_r^3 \cdot \nabla_{\theta_{\mathcal{H}^s}} (\beta_1 \cdot \mathcal{L}_{dist}^{hide} + \beta_2 \cdot \mathcal{L}_{dist}^{mia})$
        $\theta_{\mathcal{M}} \leftarrow \theta_{\mathcal{M}} - \ell_r^3 \cdot \nabla_{\theta_{\mathcal{M}}} (\beta_2 \cdot \mathcal{L}_{dist}^{mia} + \beta_3 \cdot \mathcal{L}_{dist}^{recover})$
        $\theta_{\mathcal{R}^s} \leftarrow \theta_{\mathcal{R}^s} - \ell_r^3 \cdot \nabla_{\theta_{\mathcal{R}^s}} (\beta_3 \cdot \mathcal{L}_{dist}^{recover})$
      **end**
    **end**
  **end**
**end**

---

phase, to maximize the MIA performance while distilling, we directly use the MIA label $y$ for supervision. This approach aims to ensure that the student network not only mimics the teacher's functionality closely but also enhances its capability for medical image analysis through direct guidance from the true labels.

$$
\begin{aligned}
\mathcal{L}_{dist}^{hide} &= \ell_2(x_{stego}^s, x_{stego}^t) + \sum_{i=1}^3 \ell_2(x_{stego}^{t_i}, x_{stego}^t), \\
\mathcal{L}_{dist}^{mia} &= \ell_2(x_{mia}^s, x_{mia}^t) + \sum_{i=1}^3 \ell_2(x_{mia}^{t_i}, x_{mia}^t), \\
\mathcal{L}_{dist}^{recover} &= \mathcal{L}_{mia}(\hat{y}^s, y) + \sum_{i=1}^3 \mathcal{L}_{mia}(\hat{y}^{t_i}, y), \\
\mathcal{L}_{dist} &= \alpha_1 \cdot \mathcal{L}_{dist}^{hide} + \alpha_2 \cdot \mathcal{L}_{dist}^{mia} + \alpha_3 \cdot \mathcal{L}_{dist}^{recover},
\end{aligned}
\tag{7}
$$

where $\alpha_1, \alpha_2, \alpha_3$ are used to trade-off among these loss functions.

### 3.4 Training Process

We use a multi-stage training strategy to train HideMIA (see Algorithm 1). In the first stage, we freeze the pretrained DIH network and train the MIA network with $\mathcal{L}_{total}$. In the second stage, we freeze the pretrained DIH network and MIA network to prune the LightIH with $\mathcal{L}_{dist}$. Finally, we use $\mathcal{L}_{total}$ to supervise the fine-tuning of the whole HideMIA, i.e., MIA network and LightIH. Note that $\mathcal{L}_{dice}$ is not included for the classification task.

$$
\begin{aligned}
\mathcal{L}_{mia} &= \mathcal{L}_{ce}(\hat{y}, y) + \mathcal{L}_{dice}(\hat{y}, y), \quad \mathcal{L}_{percept} = \ell_2(x_{mia}, x_{cover}), \\
\mathcal{L}_{total} &= \beta_1 \cdot \mathcal{L}_{mia} + \beta_2 \cdot \mathcal{L}_{percept},
\end{aligned}
\tag{8}
$$

where $\beta_1, \beta_2$ are used to trade-off between $\mathcal{L}_{mia}$ and $\mathcal{L}_{percept}$.

## 4 Experiments and Results

### 4.1 Experimental Setup

**Datasets.** For comprehensive comparisons, we use six widely used MIA datasets, including three segmentation datasets and three classification datasets. It should be noted that all selected datasets are acquired with different imaging devices and capture distinct subjects. These datasets are BUSI (breast ultrasound tumor segmentation with 612 images) [1], Kvasir-SEG (endoscopic polyp segmentation with 1,000 images) [16], ChildDental (tooth X-ray image segmentation with 2,489 images) [5, 15], SIPaKMed [42] (pathological cervical cell classification with 1,510 images), DermaMNIST [54] (dermatoscopic skin lesion classification with 10,015 images), and ChestCT [41] (chest computed tomography image classification with 1,000 images). For all experiments in this work, we follow the official splitting configuration of the datasets. For those without an official configuration (BUSI, Kvasir-SEG, and SIPaKMed), we randomly divide the data into training and testing sets at a ratio of 8:2. We randomly sample 512 images of MS COCO [28] as the cover image dataset. There is no overlap between the training cover images and testing cover images.

**Competing Methods.** To the best of our knowledge, no method is proposed to perform MIA in the steganographic domain. To ensure a comprehensive and fair comparison, we select three widely used DIH methods (i.e., DeepStega [4], HiDDeN [65], and HiNet [17]) and four famous or SOTA medical image segmentation networks, i.e., U-Net [44], TransUNet [7], XNet [61], and CMUNeXt [47]. Then we pair each DIH method with each segmentation network, forming twelve competing methods. For classification tasks, we also employ a pair of "segmentation network + DIH network" for comparison. This is due to the architecture of our HideMIA, where the MIA network is designed with an encoder-decoder structure. Employing segmentation networks, which also use this encoder-decoder structure, rather than encoder-only structured classification networks, allows fairer comparisons.

**Evaluation Metrics.** Dice Similarity Coefficient (DSC) and Average Surface Distance (ASD) are adopted for medical image segmentation tasks [43]. The medical image classification performance is evaluated by Accuracy (Acc) and Area Under receiver operating characteristic Curve (AUC). We calculate the average Peak Signal-to-Noise Ratio (PSNR) and Structural Similarity Index (SSIM) between $x_{mia}$ and $x_{cover}$ to assess the imperceptibility. We randomly sample five cover images to calculate the mean values of these metrics.

**Implementation Details.** The size of the images is standardized to 224×224. We use the Adam optimizer with learning rates $\ell_r^1, \ell_r^2, \ell_r^3$ of $5\times10^{-4}, 1\times10^{-4}, 1\times10^{-5}$, respectively. The batch size is 8. During training, all cover images $x_{stego}$ are randomly selected. In the loss functions, i.e., Eqs. (7)-(8), the weights $\alpha_1, \alpha_2, \alpha_3, \beta_1$, and $\beta_2$ are respectively set to 800, 800, 4, 1, and 200. We set $\lambda$ of MoDC shown in Eq. (3) to 0.5. For the secret images, $x_{secret}$, hiding is performed across three channels, and recovery for $x_{res}$ is also conducted across three channels. During segmentation, the recovered $x_{res}$ has its three channels averaged followed by a sigmoid activation to obtain the final binary map. For classification tasks, a classification head is appended after the recovered results to obtain the predictions.

**Table 1: Comparisons of covert medical image segmentation and classification performance across six datasets. DSC (%) and ASD are reported for segmentation datasets, while Acc (%) and AUC (%) are reported for classification datasets.**

| Task → | | Segmentation | | | | | | Classification | | | | | |
|---|---|---|---|---|---|---|---|---|---|---|---|---|---|
| Dataset → | | BUSI | | Kvasir-SEG | | ChildDental | | SIPaKMeD | | DermaMNIST | | ChestCT | |
| MIA ↓ | Hiding ↓ | DSC ↑ | ASD ↓ | DSC ↑ | ASD ↓ | DSC ↑ | ASD ↓ | Acc ↑ | AUC ↑ | Acc ↑ | AUC ↑ | Acc ↑ | AUC ↑ |
| U-Net | - | 72.85 | 6.58 | 81.83 | 4.30 | 91.15 | 0.15 | - | - | - | - | - | - |
| ResNet50 | - | - | - | - | - | - | - | 67.23 | 84.35 | 73.10 | 91.20 | 58.41 | 85.37 |
| U-Net | HiNet | 60.55 | 13.76 | 64.82 | 9.47 | 77.62 | 1.18 | 44.63 | 73.74 | 67.08 | 82.72 | 31.75 | 69.60 |
| TransUNet | HiNet | 63.92 | 10.61 | 60.29 | 11.5 | 74.35 | 1.42 | 49.72 | 81.07 | 68.33 | 77.10 | 38.10 | 55.04 |
| XNet | HiNet | 59.74 | 13.43 | 61.60 | 10.88 | 74.66 | 1.49 | 44.41 | 75.07 | 69.23 | 86.46 | 31.43 | 69.43 |
| CMUNeXt | HiNet | 64.36 | 11.04 | 68.26 | 9.15 | 75.23 | 1.78 | 47.46 | 79.93 | 67.48 | 86.47 | 33.65 | 65.19 |
| U-Net | DeepStega | 66.77 | 9.45 | 60.38 | 8.42 | 69.84 | 2.11 | 43.05 | 79.03 | 66.18 | 77.97 | 33.33 | 54.44 |
| TransUNet | DeepStega | 62.16 | 11.95 | 58.93 | 13.11 | 71.54 | 1.43 | 47.91 | 83.92 | 65.59 | 85.50 | 34.29 | 64.93 |
| XNet | DeepStega | 65.44 | 9.65 | 63.71 | 11.53 | 71.12 | 1.57 | 56.72 | 85.99 | 66.38 | 82.70 | 35.24 | 66.05 |
| CMUNeXt | DeepStega | 61.41 | 12.93 | 65.23 | 10.22 | 76.08 | 1.80 | 52.54 | 76.57 | 67.23 | 85.40 | 33.02 | 69.92 |
| U-Net | HiDDeN | 61.79 | 10.38 | 65.92 | 8.67 | 74.90 | 2.65 | 38.98 | 75.96 | 67.13 | 80.33 | 30.16 | 65.32 |
| TransUNet | HiDDeN | 61.68 | 10.54 | 61.82 | 12.83 | 77.39 | 1.73 | 49.94 | 82.65 | 68.28 | 78.97 | 29.84 | 66.08 |
| XNet | HiDDeN | 63.45 | 12.09 | 63.83 | 11.88 | 70.07 | 1.49 | 44.18 | 74.31 | 66.88 | 72.94 | 31.11 | 65.41 |
| CMUNeXt | HiDDeN | 63.29 | 12.95 | 64.51 | 11.98 | 78.59 | 1.1 | 42.37 | 73.79 | 67.68 | 81.88 | 37.14 | 54.24 |
| **HideMIA (Ours)** | | **74.11** | **6.76** | **77.56** | **5.79** | **89.45** | **0.19** | **63.28** | **87.25** | **78.10** | **92.73** | **51.75** | **72.72** |

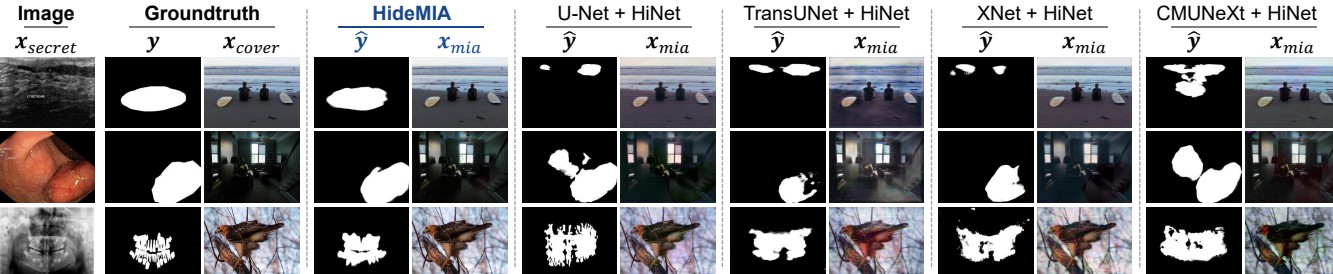

**Figure 4: Visual comparisons for covert medical image segmentation on BUSI, Kvasir-SEG, and ChildDental.**

**Table 2: Comparisons of imperceptibility of the covert MIA. We report average PSNR and SSIM between $x_{mia}$ and $x_{cover}$ on all segmentation datasets and classification datasets.**

| Method | | Segmentation | | Classification | |
|---|---|---|---|---|---|
| MIA | Hiding | PSNR ↑ | SSIM ↑ | PSNR ↑ | SSIM ↑ |
| U-Net | HiNet | 18.58 | 0.2953 | 19.29 | 0.3460 |
| TransUNet | HiNet | 17.35 | 0.3251 | 20.30 | 0.3694 |
| XNet | HiNet | 18.69 | 0.2904 | 17.42 | 0.2473 |
| CMUNeXt | HiNet | 18.72 | 0.2916 | 19.77 | 0.3850 |
| U-Net | DeepStega | 18.07 | 0.3150 | 21.49 | 0.3759 |
| TransUNet | DeepStega | 17.38 | 0.3653 | 15.85 | 0.2721 |
| XNet | DeepStega | 18.11 | 0.4131 | 19.57 | 0.3561 |
| CMUNeXt | DeepStega | 18.07 | 0.3678 | 20.04 | 0.3697 |
| U-Net | HiDDeN | 17.08 | 0.2865 | 17.03 | 0.2980 |
| TransUNet | HiDDeN | 16.83 | 0.2217 | 16.43 | 0.2759 |
| XNet | HiDDeN | 16.60 | 0.2387 | 16.30 | 0.2706 |
| CMUNeXt | HiDDeN | 18.13 | 0.2746 | 13.48 | 0.4532 |
| **HideMIA (Ours)** | | **33.01** | **0.8556** | **36.84** | **0.8963** |

## 4.2 Comparison Results

**Segmentation and Classification Results.** As shown in Table 1 and Fig. 4, HideMIA achieves the highest segmentation and classification performance across three datasets compared to other covert MIA methods, respectively. On average, its segmentation performance surpasses that of the second-place *CMUNeXt + HiNet* by 11.09% in DSC and is lower by 3.07 in ASD. When compared to the vanilla framework (without privacy-enhancing) utilizing U-Net, HideMIA's average performance only shows a slight decrease of 1.57% in DSC, while ASD increases by 0.57. Meanwhile, its classification performance surpasses that of the second-place *XNet +*

*DeepStega* by 11.60% in Acc and 5.98% in AUC. When compared to the vanilla framework (without privacy-enhancing) utilizing U-Net, HideMIA's average performance only shows a slight decrease of 1.87% in Acc and 2.74% in AUC. This indicates HideMIA's effectiveness in maintaining high MIA performance (both for segmentation and classification) while enhancing privacy protection. We note that XNet is also based on wavelet transforms, similar to HideMIA, underperforms due to its lack of adaptive analysis across different bands, unlike HideMIA. Additionally, despite TransUNet, XNet, and CMUNeXt having incremental designs over U-Net, their performance does not improve across all datasets. This inconsistency can be attributed to the fact that their incremental designs are not effective in the steganographic domain.

**Imperceptiblity.** Results in Table 2 demonstrate that our HideMIA outperforms competing methods in PSNR and SSIM for both classification and segmentation tasks, with a significant improvement over other methods. As shown in Fig. 4, $x_{mia}$ generated by HideMIA has the highest visual similarity to $x_{cover}$. Compared to $x_{cover}$, there are significant color modifications and obvious horizontal striping artifacts in $x_{mia}$ generated by other methods. The poor imperceptibility of competing methods' $x_{mia}$ is attributed to the absence of low-frequency guidance from $x_{stego}$. Results in Table 3 demonstrate that removing $x_{stego}^{LL}$ in Eq. (1) significantly decreases PSNR and SSIM, yet the MIA performance remains almost unaffected. The imperceptibility of $x_{stego}$ will be discussed in Sec. 4.3.

**Table 3: Ablation results on the LL-band residual ($x_{stego}^{LL}$) in Eq. (1) over BUSI.**

| Residual of $x_{stego}^{LL}$ → | w/ | w/o |
|---|---|---|
| PSNR ↑ | **33.84** | 17.41 |
| SSIM (%) ↑ | **83.94** | 26.15 |
| DSC (%) ↑ | 74.11 | **74.39** |
| ASD ↓ | **6.76** | 6.89 |

**Table 4: Ablation results on different numbers of activated difference convolutions within MoDC over BUSI.**

| Activated Number $k$ | DSC (%) ↑ | ASD ↓ |
|---|---|---|
| 0 (Vanilla Conv.) | 70.77 | 9.91 |
| 1 | 72.75 | 9.12 |
| **2** | **74.11** | **6.76** |
| 3 | 73.47 | 7.01 |
| 4 | 71.88 | 7.34 |
| 5 | 71.37 | 7.36 |

**Table 5: Ablation results on utilizing different difference convolutions within MoDC over BUSI. We fix the activated number $k$ to 2.**

| Candidate Set of DCs | DSC (%) ↑ | ASD ↓ |
|---|---|---|
| $\{\mathcal{D}_1, \mathcal{D}_2\}$ | 70.47 | 8.91 |
| $\{\mathcal{D}_1, \mathcal{D}_2, \mathcal{D}_3\}$ | 71.51 | 8.73 |
| $\{\mathcal{D}_1, \mathcal{D}_2, \mathcal{D}_3, \mathcal{D}_4\}$ | 73.80 | 6.89 |
| $\{\mathcal{D}_1, \mathcal{D}_2, \mathcal{D}_3, \mathcal{D}_4, \mathcal{D}_5\}$ | **74.11** | **6.76** |

**Table 6: Ablation results on AsyWA. We compare different variations of AsyWA over BUSI.**

| Variation of AsyWA | DSC (%) ↑ | ASD ↓ |
|---|---|---|
| w/o attention | 68.94 | 10.14 |
| Inner-band | 69.91 | 8.22 |
| Inter-band (Symmetry) | 71.35 | 7.39 |
| Inter-band (Asymmetric) | 73.38 | 6.75 |
| Inner- & Inter-band (Symmetry) | 72.32 | 9.36 |
| **Inner- & Inter-band (Asymmetric)** | 74.11 | 6.76 |

**Table 7: Ablation results on LightIH. We calculate PSNR and SSIM between $x_{stego}$ and $x_{cover}$ to evaluate hiding performance over BUSI.**

| DIH Network | FLOPs ↓ | Parameters ↓ | DSC (%) ↑ | PSNR ↑ | SSIM ↑ |
|---|---|---|---|---|---|
| HiNet | 152.17G | 4.05M | 71.45 | **45.27** | **0.989** |
| DeepStega | 73.30G | **0.49M** | 70.25 | 31.76 | 0.812 |
| HiDDeN | 87.49G | 0.58M | 69.80 | 34.16 | 0.969 |
| LightIH (Feature-Aligned) | **38.04G** | 1.01M | 72.06 | 41.83 | 0.975 |
| **LightIH (Function-Aligned)** | **38.04G** | 1.01M | **74.11** | 43.60 | 0.982 |

## 4.3 Ablation Study

**Effectiveness of MoDC.** Results in Table 4 reveal that when the activation number is 0 (using vanilla convolution), there is a significant performance decline compared to others. This verifies the effectiveness of the designed difference convolutions in the steganographic domain. Moreover, we observe that the performance peaks when the activation number is 2, indicating that too few activated DCs fail to form convolution operations tailored to the steganographic analysis suitable for the respective band, leading to inadequate MIA feature extraction. Conversely, too many activations introduce inappropriate kernels, introducing noise into the MIA process. Besides, we fix the number of activations to 2, which shows the best MIA performance, and compare the performance under various sets of candidate DCs in Table 5. When the candidate set includes the full set, *i.e.* $\{\mathcal{D}_1, \mathcal{D}_2, \mathcal{D}_3, \mathcal{D}_4, \mathcal{D}_5\}$, the performance is optimal. A candidate DC count of 4 outperforms a count of 3, and similarly, a count of 3 outperforms a count of 2. These results validate the effectiveness of each of the novel DCs we propose.

**Effectiveness of AysWA.** As shown in Table 6, performance is at its lowest without any attention mechanism, indicating that global perception across spatial and spectral dimensions is crucial for MIA in the steganographic domain. Besides, integrating only inner-band or inter-band attention is less effective than having both, further validating the effectiveness of the proposed cross-spatial and cross-spectral interactions. Meanwhile, we observe that inter-band (Asymmetric) outperforms Inter-band (Symmetry) and Inner- & Inter-band (Asymmetric) outperforms Inner- & Inter-band (Symmetry). This demonstrates that our asymmetric design for the LL band is effective for the HideMIA framework. It can help resist the noisy cover-specific information in the LL band.

**Effectiveness of LightIH.** We compare LightIH, HiNet, DeepStega, and HiDDeN with resource metrics, *i.e.,* Floating Point Operations (FLOPs) and Parameters, image hiding performance (PSNR and SSIM between $x_{stego}$ and $x_{cover}$), and MIA performance (DSC). As shown in Table 7, since having a quarter of the INN layers compared to HiNet, LightIH exhibits lower FLOPs than other DIH networks. LightIH has fewer parameters than its teacher (HiNet),

slightly more than DeepStega and HiDDeN. However, due to our function-aligned distillation strategy, LightIH significantly outperforms DeepStega and HiDDeN in image hiding performance, with a substantial advantage in MIA performance as well. In real-world scenarios, a PSNR over 40 is already very difficult for human eyes to distinguish. Therefore, trading off a slight decrease in imperceptibility for reduced resource consumption and improved MIA performance compared to the teacher network, HiNet, is worthwhile. Besides, we also notice that aligning features directly is not as effective as aligning the DIH and MIA functions in a multi-stage framework like HideMIA.

## 5 Conclusion

We propose a covert client-server MIA framework that effectively defends against attacks from both the transmission channel and malicious third-party servers. HideMIA is less likely to arouse suspicion among attackers compared to existing privacy-enhancing MIA methods. Comprehensive experiments demonstrate that within HideMIA, our proposed MoDC and AsyWA enable more effective covert analysis directly in the steganographic domain. Additionally, our LightIH obtained by function-aligned knowledge distillation facilitates the deployment of a lightweight DIH network on the client side, making HideMIA more practical. HideMIA achieves the SOTA MIA performance and imperceptibility in medical image segmentation and image classification. It also brings insight into fine-grained analysis in the steganographic domain.

However, as this is the first work to protect the client-server MIA framework through image hiding, there are some limitations that we hope to address in the future: (1) *Better imperceptibility*. As illustrated in Tables 2 and 7, the imperceptibility of $x_{mia}$ is lower than that of $x_{stego}$. After processing through the MIA network, there are usually some stripe-like artifacts (see Fig. 4). Eliminating these artifacts would make attackers more difficult to detect. (2) *Lossless MIA*. Compared to non-privacy-enhancing frameworks, there is still a gap in MIA performance, indicating that the distortion introduced by DIH brings side effects. Designing a nearly lossless covert MIA framework is worth exploring.

## Acknowledgement

This research was done at the Rapid-Rich Object Search (ROSE) Lab, Nanyang Technological University, and supported in part by the National Natural Science Foundation of China (Grants No. 62272022, 62306061, 62276254, and U23B2054), the National Key Research and Development Program of China (Grants No. 2022YFB3207700 and 210YBXM2024106007), Guangdong Basic and Applied Basic Research Foundation (Grant No. 2023A1515140037), the InnoHK program, and the NTU-PKU Joint Research Institute (sponsored by the Ng Teng Fong Charitable Foundation).

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
