# OpenReview forum: "HideMIA: Hidden Wavelet Mining for Privacy-Enhancing Medical Image Analysis"
_acmmm.org/ACMMM/2024/Conference — MM2024 Poster_

### Official Review · Reviewer_1Zr5 · 2024-05-23

**Rating:** 3
**Confidence:** 4

**Summary:**

This paper introduces a covert Medical Image Analysis (MIA) framework based on deep image hiding, namely HideMIA, where the Mixture-of-Difference-Convolutions (MoDC) and Asymmetric Wavelet Attention (AsyWA) are proposed to enable HideMIA to conduct fine-grained analysis on each wavelet sub-band within the steganographic domain, mining features that are irrelevant to the cover and specific to medical images.

**Strengths:**

1. The proposed method is effective to a certain extent and the paper is well-written, which makes it easier to understand the methodology proposed by the authors.
2. The experimental design is reasonable and convincing. Ablation experiments are conducted specifically targeting the innovative points.

**Limitations:**

1. It is well known that the stego images in the steganographic domain can resist steanalyzers. However, it seems from Figure 1 that the authors did not consider this.
2. For the proposed HideMIA, it can resist the member inference attacks? If possible, what is the underlying cause?
3. As far as I know, the data format of medical images is different from that of natural images in practice. Therefore, did the author consider a customized network design for data loading?
4. How to determine the weight value of each sub-loss item in Equation 7?
5. In the comparison results, the author needs to consider how the adopted image hiding strategy will affect the recognition accuracy.
6. Please recheck the manuscript, where there are some grammatical errors.

**Suitability:**

2

---

### Official Review · Reviewer_Cb5x · 2024-05-25

**Rating:** 4
**Confidence:** 3

**Summary:**

The idea is to use image steganography in training medical image models. Authors suggest a technique called HideMIA (deep image hiding), where medical images can be hidden inside any natural cover image, which makes it difficult for intermediaries to understand that model under training. HideMIA consists of an image hiding network, medical image analysis network (MIA), and image recovering network.
The image hiding and image recovering networks are deployed on the client side, while the medical image analysis network operates on the server.

Note:  Abstract is clear and readable. There is scope for improvement in last 2 sentences with specifics of comparison on how the proposed method is superior with at least one rubric.

**Strengths:**

1. Novelty: The paper addresses a significant challenge in MIA by proposing a covert framework for protecting medical image privacy.

2. Theoretical Approach: Innovative techniques like Mixture-of-Difference-Convolutions (MoDC) and Asymmetric Wavelet Attention (AsyWA) and Function-Aligned Distillation are introduced, enhancing security while minimizing detection risk.

3. Technical correctness: Paper presents detailed mathematical equations which are well described and formatted, including the algorithms mentioned in the paper.

4. Evaluation: Comprehensive experiments on six datasets demonstrate superior performance of HideMIA in both segmentation and classification tasks, surpassing existing methods. Ablation study is conducted in detail.

**Limitations:**

1. Model is likely to underperform with training techniques used in Federated learning as they may be more robust and secure. No such comparison is found in paper with alternative methods that may outperform. At least the limited scope assumption should be mentioned.

2. Though discussion section provides limitation and the future scope, specifics are missing and some of the assumptions in synthetic medical cover images itself is contradictory to the purpose of the solution in the paper.

3. Training Process (3.4 section) needs to be expanded to gain more clarity on the steps involved.

4. Comparison of the results (4.2 section) needs to me more detailed for better understating and readability.

**Suitability:**

3

---

### Official Review · Reviewer_orSB · 2024-05-25

**Rating:** 5
**Confidence:** 4

**Summary:**

This paper proposes a framework for medical image steganalysis, HideMIA, which proposes Mixture-of-Difference-Convolutions (MoDC) and Asymmetric Wavelet Attention (AsyWA), enabling HideMIA to perform analysis on each wavelet subband in the steganographic domain. Perform fine-grained analysis to mine carrier image-independent and medical image-specific features. Moreover, in order to reduce the resource consumption of HideMIA on the client device, this paper designs functionally aligned knowledge distillation to obtain a lightweight image hiding network, namely Light IH. This paper is the first work to hide medical images and directly perform medical image analysis on the hidden images. It is also the first work to apply knowledge distillation to develop a lightweight deep image hiding network.

**Strengths:**

1. The paper has a high degree of novelty. It applies deep image hiding technology to medical image hiding and analysis for the first time, and will develop a lightweight deep image hiding network through knowledge distillation.
2. In Section 3 of the paper, the specific structure of the proposed HideMIA framework is described clearly, and the technical details (such as network structure and training methods) of the proposed MoDC and AsyWA are also very clear.
3. The paper conducts a detailed experimental evaluation, and conducts comparative experiments and ablation experiments on six data sets. The model selection and design for comparison are more reasonable and sufficient, and the display and analysis of the experimental results are also relatively clear, such as the image of Fig4 result.

**Limitations:**

There are some minor problems in the presentation details of the paper, such as an error in the reference to Figure 3 in Section 3.3.

**Suitability:**

2

---

### Meta-Review · Area_Chair_ceLG · 2024-07-03

**Recommendation:** Accept (Poster)
**Confidence:** 5

**Metareview:**

Two reviewers are positive and one reviewer is slightly negative towards this work. As membership inference attack is a bit beyond the scope of this work, the discussions on this type of attack can be given in authors' future work. But the authors are recommended to proofread the paper for the final version.